# A Bilingual, Open World Video Text Dataset and End-to-end Video Text Spotter with Transformer

**Weijia Wu**[*†]
Zhejiang University
`weijiawu@zju.edu.cn`

**Yuanqiang Cai**[*]
Beijing University of Posts and Telecommunications
`caiyuanqiang@bupt.edu.cn`

**Debing Zhang**
Kuaishou Technology

**Sibo Wang**
Kuaishou Technology

**Zhuang Li**
Kuaishou Technology

**Jiahong Li**
Kuaishou Technology

**Yejun Tang**
Kuaishou Technology

**Hong Zhou**[‡]
Zhejiang University

## Abstract

Most existing video text spotting benchmarks focus on evaluating a single language and scenario with limited data. In this work, we introduce a large-scale, **B**ilingual, **O**pen World **V**ideo text benchmark dataset(BOVText). There are four features for BOVText. Firstly, we provide **2,000+** videos with more than **1,750,000+** frames, **25** times larger than the existing largest dataset with incidental text in videos. Secondly, our dataset covers **30+** open categories with a wide selection of various scenarios, *e.g., Life Vlog, Driving, Movie, etc*. Thirdly, abundant text types annotation (*i.e., title, caption or scene text*) are provided for the different representational meanings in video. Fourthly, the BOVText provides bilingual text annotation to promote multiple cultures' live and communication.

Besides, we propose an end-to-end video text spotting framework with Transformer, termed TransVTSpotter, which solves the multi-orient text spotting in video with a simple, but efficient attention-based query-key mechanism. It applies object features from the previous frame as a tracking query for the current frame and introduces a rotation angle prediction to fit the multi-orient text instance. On ICDAR2015(video), TransVTSpotter achieves the state-of-the-art performance with **44.1%** MOTA, **9** fps. The dataset and code of TransVTSpotter can be found at `github.com/weijiawu/BOVText` and `github.com/weijiawu/TransVTSpotter`, respectively.

## 1 Introduction

Text spotting [24, 15] has received increasing attention due to its numerous applications in computer vision, *e.g.,* document analysis, image-based translation, image retrieval [37, 29], etc. With the advent of deep learning and abundance in digital data, reading text from static images has made extraordinary progress in recent years with a lot of great public datasets [11, 16, 6] and algorithms [48, 62, 27, 23]. By contrast, video text spotting almost remains at a standstill for the lack of large-scale multidimensional practical datasets, which limited numerous applications of video text, *e.g.,* video understanding [40], video retrieval [8], video text translation, and license plate recognition [1], etc.

---

[*]Equal contribution.

[†]This work was done when Weijia Wu were interns in MMU, Kuaishou Technology, Beijing, China.

[‡]Corresponding author.

35th Conference on Neural Information Processing Systems (NeurIPS 2021) Track on Datasets and Benchmarks.

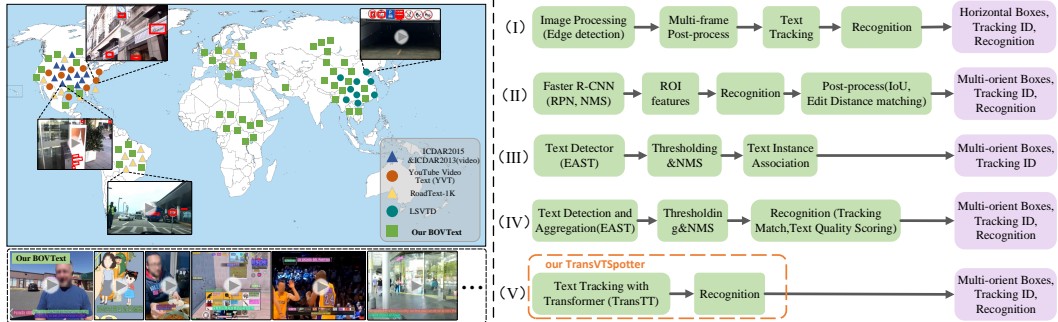

| | | | | | | Horizontal Boxes, |
|---|---|---|---|---|---|---|
(I) Image Processing (Edge detection) → Multi-frame Post-process → Text Tracking → Recognition → Horizontal Boxes, Tracking ID, Recognition

(II) Faster R-CNN (RPN, NMS) → ROI features → Recognition → Post-process(IoU, Edit Distance matching) → Multi-orient Boxes, Tracking ID, Recognition

(III) Text Detector (EAST) → Thresholding &NMS → Text Instance Association → Multi-orient Boxes, Tracking ID

(IV) Text Detection and Aggregation(EAST) → Thresholding&NMS → Recognition (Tracking Match,Text Quality Scoring) → Multi-orient Boxes, Tracking ID, Recognition

our TransVTSpotter
(V) Text Tracking with Transformer (TransTT) → Recognition → Multi-orient Boxes, Tracking ID, Recognition

（a）Comparison of different data distributions    （b）Comparison of different pipelines for video text spotting

Figure 1: **Comparison of dataset distribution and pipeline**. (a) Data distribution. BOVText provides various open world scenarios with unique *NBA, Game, etc.* (b) Pipelines: (I) Multi-stage pipeline in [52], [13], [54] *etc*; (II) Multi-orient video text spotting pipeline with Faster R-CNN [33] proposed by Wang *et al.* [49]; (III) Online text tracking pipeline in Yu *et al.* [57]; (IV) Fast video text spotting pipeline in Cheng *et al.* [5]; (V) An end-to-end pipeline with transformer in this work.

Video text spotting(VTS) is the task that requires simultaneously classifying, detecting, tracking and recognizing text instances in a video sequence. There have been a few previous works [53, 51] and datasets [31, 17] in the community for attempting to develop video text spotting. ICDAR2015 (Text in Videos) [16] was introduced during the ICDAR Robust Reading Competition in 2015 and mainly includes a training set of 25 videos (13k frames) and a test set of 24 videos (14k frames). The videos were categorized into seven scenarios: walking outdoors, searching for a shop in a shopping street, etc. YouTube Video Text (YVT) [31] dataset harvested from YouTube, contains 30 videos with 13k frames, 15k for training, and 15k for testing. The text content in the dataset mainly includes overlay text and scene text (*e.g.,* street signs, business signs, words on shirt). RoadText-1K [32] are sampled from BDD100K [56], includes 700 videos (210k frames) for training and 300 videos for testing. The texts are all obtained from driving videos and match for driver assistance and self-driving systems. LSVTD [5] includes 100 text videos, 13 indoor (*e.g.,* bookstore, shopping mall) and 9 outdoor (*e.g.,* highway, city road) scenarios. However, as shown in Figure. 1 (a), most existing video text datasets are limited by the amount of training data (less than 300k frames), single video scenarios, and a single language. There are only a few outdoor scene text videos with 13k frames in ICDAR2015 (video). Similar situation for YVT, RoadText-1k, and LSVTD, the training set is limited and the dataset scenarios are single. This makes it difficult to evaluate the effectiveness of more advanced deep learning models for more open scenarios, such as *game, sport and news report*. Besides, most existing video text datasets are proposed before 2019 years, and some of them are no longer being maintained without an open-source evaluation script. The download links of YVT even have become invalid, which is not conducive to the development of video text spotting.

In this work, we contribute a large-scale, bilingual open-world benchmark dataset (BOVText) to the community for developing and testing video text spotting that can fare in a realistic setting. Our dataset has several advantages. **Firstly**, the large training set (*i.e.,* 2,000+video and 1,750,000+ video frames) from *KuaiShou* and *YouTube* enables the development of deep design specific for video text spotting. **Secondly**, unlike the existing datasets, BOVText support **30+** open scenarios, including many new scenarios such as *Sportscast(NBA, FIFA World Cup...), Life Vlog, Game, etc*, as shown in Figure. 1 (a). These data is collected from the worldwide user of *YouTube*[4] and *KuaiShou*[5], cover various daily scenarios without region limitation and virtual scenes. But the previous video text datasets usually are collected toward a special city or language from the hand-held camcorder. **Thirdly**, BOVText is the first benchmark for supporting abundant text types annotation. Caption, title, and scene text are separately tagged for the different representational meanings in the video. This made our BOVText has the potential to promote other video-and-language tasks, such as video understanding. **Fourthly,** bilingual text annotation(*i.e.,* Chinese, English) is provided in BOVText to promote multiple cultures' live and communication.

Except for the promising benchmark, we also proposed a simple, but effective video text spotter with transformer (TransVTSpotter). As shown in Figure. 1 (b), unlike previous methods that involve multiple steps, such as proposal generation, text aggregation, and post-processing(NMS), TransVTSpotter

[4]https://www.youtube.com/
[5]https://www.kuaishou.com/en

only requires two steps. 1) Text tracking: for each consecutive frame image, we obtain the multi-orient boxes tracking trajectories of text by boxes IoU matching between the predicted detection boxes [4] and the predicted tracking boxes [41], where the detection boxes are obtained by taking an object query as input, just like DETR [4]. And features from previously detected objects to form another "track query" to discover associated objects (*i.e.,* the predicted tracking boxes) on the current frames. Besides, an additional angle loss of multi-orient box and *Hungarian angle cost* are design to obtain the angle of multi-orient. 2) Text recognition: recognizing the tracked texts with attention-based text recognizer [26]. Without bells and whistles, TransVTSpotter achieves state-of-the-art performance on ICDAR2015 with **44.1%** MOTA, **9** fps. The main contributions of this work are three folds:

(1) We propose a large-scale, bilingual and open world video text spotting benchmark named BOVText. The proposed dataset provides **2,000+** videos, **1,750,000** frames, open videos scenarios (*e.g., Indoor, Outdoor, Game, Sport*), abundant text types (*i.e., title, caption or scene text*), multi-stage tasks and is **25** times the existing largest dataset with incidental text.

(2) Caption, title, scene text, and other overlap texts are firstly separately tagged for the different representational meanings in the video. Based on the previous works [37, 19], this favors other tasks theoretically, such as video understanding, video retrieval, and video text translation.

(3) We first propose a new video text spotting framework with Transformer, termed **TransVTSpotter**, which solves the video multi-orient text spotting with a simple, but effective pipeline based on the tracking query-key mechanism and rotated boxes angle prediction.

(4) We evaluate and compare TransVTSpotter and other techniques for scene text detection, recognition, text tracking, and end-to-end video text spotting on BOVText and other existing datasets. Besides, a thorough analysis of performance on the proposed dataset is provided.

## 2 Related Work

### 2.1 End-to-End Text Spotting

For image-level text spotting, various methods [20, 12, 27] based on deep learning have been proposed and have improved the performance considerably. Li *et al.* [20] proposed the first end-to-end trainable scene text spotting method. The method successfully uses a RoI Pooling [33] to joint detection and recognition features. Liao *et al.* [27] propose a Mask TextSpotter which subtly refines Mask R-CNN and uses character-level supervision to detect and recognize characters simultaneously.

However, these methods based on the static image can not obtain temporal information in the video, which is essential for some downstream tasks such as video understanding. Compared to text spotting in a static image, video text spotting methods are rare. Yin *et al.* [55] provides a detailed survey, summarizes text detection, tracking and recognition methods in video. Wang *et al.* [49] introduced an end-to-end video text recognition method through associations of texts in the current frame and several previous frames to obtain final results. Cheng *et al.* [5] propose a video text spotting framework by only recognizing the localized text one-time. Nguyen *et al.* [31] improves detection and recognition performance by temporal redundancy and linearly interpolate to recover missing detection results. Rong *et al.* [36] tracked video text using tracking-by-detection. An MSER detector was used to locate scene text character, which was used as a constraint to optimize the trajectory search. To promote video text spotting, we attempt to establish a standardized benchmark (BOVText), covering various open scenarios and bilingual text annotation.

### 2.2 Text Spotting Datasets for Images and Videos

The various and practical benchmark datasets [16, 43, 17, 6, 17] contribute to the huge success of scene text detection and recognition at the image level. ICDAR2015 [16] was provided from the ICDAR2015 Robust Reading Competition. Google glasses capture these images without taking care of position, so text in the scene can be in arbitrary orientations. ICDAR2017MLT [30] is a large-scale bilingual text dataset, which is composed of complete scene images which come from 9 languages. The COCO-Text dataset [43] is currently the largest dataset for scene text detection and recognition. It contains 50,000+ images for training and testing.

The development of video text spotting is limited in recent years due to the lack of efficient data sets. ICDAR 2015 Video [17] consists of 28 videos lasting from 10 seconds to 1 minute in indoors

or outdoors scenarios. Limited videos (*i.e.,* 13 videos) used for training and 15 for testing. Minetto Dataset [28] consists of 5 videos in outdoor scenes. The frame size is 640 x 480 and all videos are used for testing. YVT [31] contains 30 videos, 15 for training and 15 for testing. Different from the above two datasets, it contains web videos except for scene videos. USTB-VidTEXT [53] with only five videos mostly contain born-digital text sourced from Youtube. RoadText-1K [32] provides a driving videos dataset with 1000 videos. The 10-second long video clips in the dataset are sampled from BDD100K [56]. As shown in Table. 1, the existing datasets contain a limited training set and single video scenarios. To promote the development of video text spotting, we create a large-scale, bilingual open-world benchmark dataset.

## 2.3 Transformers in Vision

Transformer is first proposed in [42] as a new paradigm for machine translation. But recently, there is a popularity of using transformer architecture in vision tasks, such as detection [4, 64], segmentation [61], 3D data processing [60], video object tracking [46] and even backbone construction [9]. Lately, some works [50, 59, 46] show using a transformer in processing sequential visual data also make remarkable shots. MOTR [59] introduce the concept of track query and the contiguous query passing mechanism for multiple-object Tracking. VisTR [50] solves instance segmentation by learning the pixel-level similarity and instance tracking is to learn the similarity between instances. But for video text spotting or multi-orient text tracking, to the best of our knowledge, there are still no transformer-based solutions while it is intuitive for its good capacity in temporal processing. Here, we propose the TransVTSpotter method and provide an affirmative answer to that, which shows convincingly high performance on the popular benchmark.

## 3 BOVText Benchmark

### 3.1 Data Collection and Annotation

**Data Collection.** To obtain abundant videos with various text types, we first start by acquiring a large list of different scenarios with text (*e.g., game scenario, travel scenario*) using *YouTube*[6] and *KuaiShou*[7] - an online resource that contains billions of videos with various scene text from cartoon movies to human relation. Then, we choose 31 open-domain categories with 1 unknown category, *i.e., , Game, Home, Fashion,* and *Technology*, as shown in Figure. 2 (a). With each raw video category, we first choose the video clips with text, then make two rounds of manual screening to remove the ordinary videos without scene text and caption text. As a result, we obtain $2,021$ videos with $1,620,305$ video frames, as shown in Table 1. Finally, to fair evaluation, we divide the dataset into two parts: the training set with $1,328,575$ frames from $1,541$ videos, and the testing set with $429,023$ frames from $480$ videos. As shown in Figure 2, different from the existing data sets, our dataset not only cares about scene text spotting in the real world, but also focuses on caption texts in the video. For the most part, caption text represents more global information than scene text, which is quite favorable for some downstream tasks, *e.g., video understanding, video caption translation, etc*.

**Data Annotation.** We invite a professional annotation team to label each video text with four kinds of description information: the rotated bounding box describing the location information, judging the tracking identification(ID) of the same text, identifying the content of the text information, and distinguishing the category of the caption, title or scene text. To save the annotation cost, we first sample the videos, annotate each sampled video frame, and then transform the annotation information from the sampled video frame to the unlabeled video frame by interpolation. Finally, we invite an audit team to carry out another round of annotation checks, and re-label part video frames with unqualified annotation. *For video sampling*, we use uniform sampling with a sampling frequency of 3 to sample all the videos in the dataset, and obtain the sampled video frame set. *For sampling video frame annotation*, each text instance is labeled in the same quadrilateral way as in the ICDAR2015 [63]. In addition, the text instance also will be marked with two description information: the category of the caption, title or scene, the recognition content, and the tracking ID. *For interpolation on unlabeled video frames*, each text instance is marked with tracking ID and recognition content, so we can judge whether the texts in adjacent sampling frames are the same

---

[6]https://www.youtube.com/
[7]https://www.kuaishou.com/en

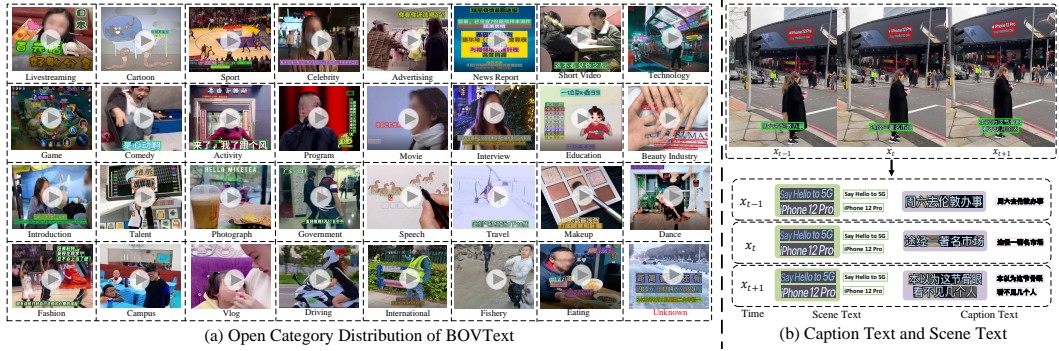

(a) Open Category Distribution of BOVText

(b) Caption Text and Scene Text

Figure 2: **Distributions of BOVText**. (a) The benchmark dataset covers a wide and open range of life scenes (32 open-domain categories). (b) Caption text (blue box) and scene text (red box) are distinguished in BOVText, which is favorable for downstream tasks.

Table 1: **Statistical Comparison.** 'D','R', 'T', 'S' and 'BI' denotes the Detection, Recognition, Tracking, Spotting and bi-lingual text, respectively. 'Incidental' denotes indoor and outdoor scenarios in daily life (*e.g., walking outdoors, driving*). 'Open' refers to any scenarios, e.g., *Game, Sport(NBA).* In green refers to these scenarios only supported by BOVText.

| Dataset | Category | BI | Task | Videos | Frames | Texts | Supported Scenario |
|---|---|---|---|---|---|---|---|
| AcTiV-D[58] | Caption | - | D | 8 | 1,843 | 5,133 | News video |
| UCAS-STLData[3] | Caption | - | D | 3 | 57,070 | 41,195 | Teleplay |
| USTB-VidTEXT[53] | Caption | - | D&S | 5 | 27,670 | 41,932 | Web video |
| YVT[31] | Scene, Caption | - | D&R&T&S | 30 | 13,500 | 16,620 | Incidental: Cartoon, Outdoor(supermarket, shopping street, driving...) |
| ICDAR2015 VT[63] | Scene | - | D&R&T&S | 51 | 27,824 | 143,588 | Incidental: Outdoor(walking, driving, supermarket, shopping street...) |
| LSVTD[5] | Scene | ✓ | D&R&T&S | 100 | 66,700 | 569,300 | Incidental: Indoor(shopping mall, supermarket, hotel...), Outdoor(driving...) |
| RoadText-1K[32] | Scene | - | D&R&T&S | 1,000 | 300,000 | 1,280,613 | Driving |
| BOVText(ours) | Scene, Caption, Title, Others | ✓ | D&R&T&S | **2,021** | **1,757,598** | **7,292,261** | Open: Cartoon, Vlog(supermarket, shopping street, driving), Travel (indoor and outdoor), Game(PUBG mobile...), Sport(NBA, world cup...), News , TV program, Education(campus, classroom, book...),Technology(introductory video, scientific propaganda...)... |

text with the same ID. For the same text instance, we first determine whether the text annotation of the sampled video frame is the starting and end frame of the text instance. If not, we look forward and backward for the starting and end position of the text instance and label it. Then we use the linear interpolation way to calculate the position of the text object in the middle of the unmarked video frame, and give tracking ID, recognition content, and category. *For check and re-label bad cases,* the linear interpolation shows a dissatisfied performance in some cases, *e.g., the new text appears on starting frame, text suddenly disappears on ending frame,* which are difficult to capture. Therefore, we invite an audit team to carry out another round of annotation checks. Around $150,000$ video frames with unqualified annotation from $1,670,305$ video frames are selected to refine, taking 20 men in three weeks. As a labor-intensive job, the whole labeling process takes **30** men in three months, *i.e.,* **21,600** man-hours, to complete about **600,000** sampled video frame annotations.

## 3.2   Dataset Analysis

The statistic comparison between BOVText and other datasets are visualized in Figure 1 (a), YouTube, and summarized in Table 1. Besides, we provide more detailed information, such as 'data distribution for 32 open scenarios', 'text language and category distribution'.

To provide the community with unified text-level quantitative descriptions, we will compare our dataset with the previous datasets from four aspects, *i.e.,* text description, video scene, dataset size, and supported tasks. *For text description attribute (i.e., Category, MLingual)*, our BOVText supports four types of text annotations(caption, title, scene, and other text) of video text with multi-language, which obviously has more extensive description ability than the existing dataset.

*For video scene attribute (i.e., Scenario)*, we present the 31 open scenarios and an "Unknown" scenarios distribution on BOVText in three levels , *i.e.,* video, video frames, and text instances , as shown in Table 2. Unlike the existing datasets, BOVText spans various video domains with

Table 2: **The Data Distribution for 32 Open Scenarios**. In green refers to these scenarios only supported by BOVText. "%" denotes the percentage of each scenario data for whole data.

| Scenarios | Video | Video Frames | Text Instances | Scenarios | Video | Video Frames | Text Instances |
|---|---|---|---|---|---|---|---|
| Cartoon | 67(3.2%) | 64,359(3.6%) | 123,191(2.1%) | Sport | 71(4.8%) | 54,643(3.1%) | 266,996(4.6%) |
| Vlog | 90(4.2%) | 83,891(4.7%) | 214,910(3.7%) | News Report | 100(4.1%) | 66,207(3.7%) | 178,000(3.1%) |
| Driving | 71(3.8%) | 61,626(3.5%) | 151,994(2.6%) | Celebrity | 50(2.4%) | 39,958(2.3%) | 121,235(2.1%) |
| Advertising | 32(1.7%) | 28,645(1.6%) | 91,090(1.0%) | Technology | 68(3.1%) | 53,305(3.1%) | 140,172(2.4%) |
| Activity | 35(1.6%) | 26,837(1.4%) | 67,879(1.2%) | Program | 42(2.3%) | 38,108(2.2%) | 214,561(3.7%) |
| Comedy | 88(4.5%) | 79,206(4.5%) | 317,865(5.5%) | Game | 21(1.0%) | 33,925(1.9%) | 84,106(1.5%) |
| Interview | 37(1.3%) | 31,440(1.8%) | 63,616(1.1%) | Livestreaming | 64(3.1%) | 62,130(3.6%) | 211,569(3.6%) |
| Government | 66(2.5%) | 45,457(2.6%) | 93,874(1.6%) | Speech | 69(3.2%) | 56,646(3.1%) | 175,119(3.0%) |
| Travel | 74(4.3%) | 71,291(4.1%) | 280,446(4.8%) | Movie | 106(5.6%) | 108,110(6.3%) | 299,760(5.2%) |
| Campus | 52(2.3%) | 43,469(2.5%) | 139,760(2.4%) | Photograph | 70(2.8%) | 64,025(3.6%) | 173,832(3.0%) |
| International | 55(3.1%) | 52,774(3.6%) | 132,117(2.3%) | Education | 74(3.4%) | 59,824(3.6%) | 360,774(6.2%) |
| Short Video | 70(4.4%) | 59,756(3.4%) | 326,930(5.6%) | Dance | 43(1.9%) | 27,941(1.6%) | 71,740(1.2%) |
| Makeup | 63(3.1%) | 54,643(3.1%) | 111,814(1.9%) | Fishery | 81(4.5%) | 75,018(4.3%) | 230,085(4.0%) |
| Talent | 86(4.1%) | 71,024(4.1%) | 339,382(5.9%) | Fashion | 63(3.0%) | 46,337(2.6%) | 98,942(1.7%) |
| Beauty Industry | 40(1.9%) | 41,350(2.4%) | 132,025(2.3%) | Introduction | 64(3.8%) | 59,048(3.4%) | 236,721(4.1%) |
| Eating | 56(2.7%) | 62,893(3.6%) | 191,035(3.3%) | Unknown | 53(2.5%) | 33,712(1.9%) | 150,721(2.6%) |

these scenarios in the existing datasets (*e.g.,* Driving for RoadText-1k [32], Vlog(supermarket, shopping street, indoor), Travel(hotel, railway station) for LSVTD [5]) and more open domains that are not yet supported (*e.g.,* Game(PUBG mobile, Honor of Kings...), Sport(NBA, world cup...), News). *For the size of the dataset(i.e., Videos, Frames, Texts)*, BOVText is **25** times larger than the existing largest dataset (*i.e.,* LSVTD [5]) with various scenarios($1, 757, 598$ v.s $66, 700$ video frames). RoadText1k [32] contains 300k videos frames, but the supported scenario is too single for only supports driving video scenarios. *For the supported tasks*, the proposed BOVText supports all video text tasks: detection, recognition, video text tracking, and end to end video text spotting. For comprehensive research, we not only focus the scale, location, recognition content, and tracking ID, but also additionally collect and annotate the category of caption, title, scene or other texts for each text instance. As shown in Figure. 2 (b), in a video, different types of text instances may exist simultaneously, and they are helpful to understand videos synergistically. Caption text can directly show the dialogue between people in video scenes and represent the time or topic of the video scenes, scene text can unambiguously define the object and can identify important localization and road paths in video scenes. Therefore, the text category annotation is favoring downstream tasks (*e.g.,* video text translation, video understanding, and video retrieval), more details in the appendix. In conclusion, the high efficiency of BOVText for evaluating advanced deep learning methods is very favorable for promoting various text spotting applications in real life. **Statistic Comparison.** As shown in Table. 1, *Category* denotes the category of the text type in the corresponding dataset. *MLingual* denotes whether the dataset contains multiple language texts. *Scenario* denotes the scene range of the video. *Videos, Frames, Texts* represents the number of videos, video frames, video texts in the dataset, respectively. *Task* denotes which tasks the dataset supports.

### 3.3 Supported Tasks and Metrics

The proposed BOVText supports four tasks: (1) Video Text Detection. (2) Video Text Recognition. (3) Video Text Tracking. (4) End to End Text Spotting in Videos.

Following ICDAR2015 [63] [8], the evaluation protocols [45] are used for text detection and recognition task. For video text tracking and spotting task, the existing video text datasets such as ICDAR2015 (video) [17] [9] and RoadText-1k [32] all adopted the MOT metrics (*i.e.,* Multiple Object Tracking Accuracy ($MOTA$) and Multiple Object Tracking Precision ($MOTP$)). However, there are two sets of measures for Multiple Object Tracking: the MOT metrics ($MOTA$,$MOTP$) [2] and ID metrics ($ID_{F1}$) [21, 35]. The CVPR19 MOTChallenge evaluation framework [7] presents that different measures serve different purposes. $Event\text{-}based$ measures like MOT help pinpoint the source of some errors and are thereby informative for the designer of certain system components. $Identity\text{-}based$ measure($ID_{F1}$) is more favorable for evaluating how well computed identities conform to true identities. Except for using MOTA, MOTP, $Identity\text{-}based$ measures($ID_{F1}$), as a new metric is adopted firstly for video text spotting task. More detailed information for metric can be obtained in the appendix.

---

[8]https://rrc.cvc.uab.es/?ch=4&com=tasks
[9]https://rrc.cvc.uab.es/?ch=3&com=tasks

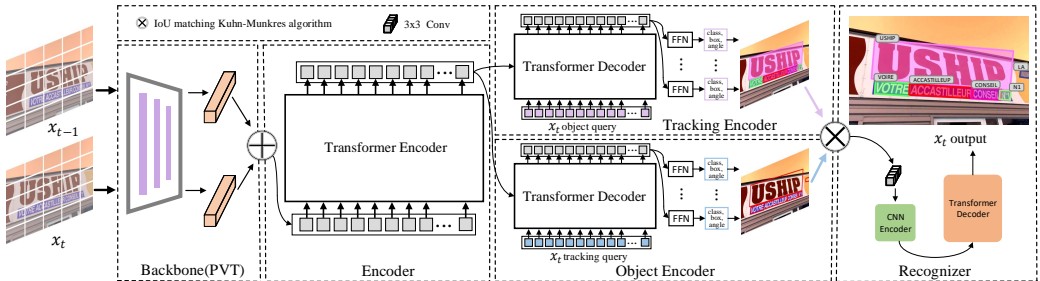

Figure 3: **Pipeline of TransVTSpotter**. It contains four main components: 1) A transformer backbone(PVT [47]) extracts feature representation of multiple images; 2) A transformer encoder models the relations of pixel-level features; 3) Two transformer decoders (shared weight) decode the instance-level features; 4) Attention-based recognizer [26] recognizes each text instance.

## 3.4 Our Method: TransVTSpotter

Two ingredients are essential for direct text spotting for TransVTSpotter: (1) A set prediction loss that forces unique matching between predicted and ground truth multi-orient boxes. (2) An architecture that predicts a set of objects and associates the same objects during different frames.

**Multi-orient Box Matching.** Compare to DETR [4], the difference is that we propose an angle prediction and corresponding loss while only horizontal boxes prediction for DETR. Let us denote the ground truth set of objects by $y$, and $\hat{y} = \{\hat{y}_i\}_{i=1}^N$ the set of $N$ predictions. $y$ is as a set of size $N$ padded with $\varnothing$ (no object). To find a bipartite matching between these two sets we search for a permutation of $N$ elements $\sigma \in \mathfrak{S}_N$ with the lowest cost:

$$\hat{\sigma} = \underset{\sigma \in \mathfrak{S}_N}{\arg\min} \sum_{i}^{N} \mathcal{L}_{\text{match}}(y_i, \hat{y}_{\sigma(i)}), \tag{1}$$

where $\mathcal{L}_{\text{match}}(y_i, \hat{y}_{\sigma(i)})$ is a pair-wise *matching cost* between ground truth $y_i$ and a prediction with index $\sigma(i)$. The matching cost takes into account the class prediction, boxes prediction and the boxes rotated angle prediction. Each element $i$ of the ground truth set can be seen as a $y_i = (c_i, b_i, a_i)$ where $c_i$ is the target class label, $b_i \in [0,1]^4$ is a vector that defines ground truth box center coordinates and its height and width relative to the image size, and $a_i$ is rotation angle between the longest edge of ground truth multi-orient box and horizontal line (x-axis). For the prediction with index $\sigma(i)$ we define probability of class $c_i$ as $\hat{p}_{\sigma(i)}(c_i)$, the predicted box as $\hat{b}_{\sigma(i)}$, and the predicted angle as $\hat{a}_{\sigma(i)}$. Thus we define $\mathcal{L}_{\text{match}}(y_i, \hat{y}_{\sigma(i)})$ as $-\mathbb{1}_{\{c_i \neq \varnothing\}}\hat{p}_{\sigma(i)}(c_i) + \mathbb{1}_{\{c_i \neq \varnothing\}}\mathcal{L}_{\text{box}}(b_i, \hat{b}_{\sigma(i)}) + \mathbb{1}_{\{c_i \neq \varnothing\}}\mathcal{L}_{\text{angle}}(a_i, \hat{a}_{\sigma(i)})$. Finally, we could compute the loss function with all pairs matched:

$$\mathcal{L}_{\text{Hungarian}}(y, \hat{y}) = \sum_{i=1}^{N} \Big[ -\log \hat{p}_{\hat{\sigma}(i)}(c_i) + \mathbb{1}_{\{c_i \neq \varnothing\}}\mathcal{L}_{\text{box}}(b_i, \hat{b}_{\hat{\sigma}}(i) + \mathbb{1}_{\{c_i \neq \varnothing\}}\mathcal{L}_{\text{angle}}(a_i, \hat{a}_{\sigma(i)})) \Big], \tag{2}$$

where $\mathcal{L}_{\text{box}}(\cdot)$ a linear combination of the $\ell_1$ loss and the generalized IoU loss [34, 4]. And $\mathcal{L}_{\text{angle}}(\cdot)$ refers to a cosine embedding loss with $1 - cos(\hat{a}_{\sigma(i)} - a_i)$.

**TransVTSpotter architecture.** The overall TransVTSpotter architecture is surprisingly simple and depicted in Figure. 3. A transformer-based backbone [47] is used to extract feature representation, transformer-based encoder-decoder framework learned current object query and previous frame tracking query as input and predicts *detection boxes* and *tracking boxes* [41]. With the detection boxes and tracking boxes, box IoU matching is used to obtain the final tracking result. Finally, attention-based recognizer [26] is utilized to obtain the final recognition results. **Text Tracking with Transformer.** Text Tracking with Transformer includes three components: backbone, transformer encoder and transformer decoder, as shown in Figure. 3. *Backbone.* Starting from the two consecutive frames $x_t \in \mathbb{R}^{3 \times H_0 \times W_0}$ and $x_{t-1} \in \mathbb{R}^{3 \times H_0 \times W_0}$, a transformer backbone [47] generates a lower-resolution activation map for the two frames($f_t \in \mathbb{R}^{C \times H \times W}$ and $f_{t-1} \in \mathbb{R}^{C \times H \times W}$), then a new feature sequence $f_t^*$ can be obtained by simple concatenating $f_t$ and $f_{t-1}$. The extracted features $f_t$ of the current frame are temporarily saved and then re-used for the next frame. *Transformer Encoder.* We adopted deformable transformer encoder [64] to model the similarities among all the pixel level features for the extracted features $f_t$. *Transformer Decoder.* Two parallel decoders [41] are employed. The object decoder takes learned object query [4] as input and predicts detection rotated boxes. The

Table 3: **Attribute Experiments for Scenarios.** Random scenarios are selected to present here.

| Method | Training Set | Detection (F-score/%) on BOVText | | | | | Tracking ($ID_{F1}$/%) on BOVText | | | | |
|--------|-------------|---------|-----------|---------|---------|------|---------|-----------|---------|---------|------|
| | | Cartoon | NewsReport | Driving | Program | Avg. | Cartoon | NewsReport | Driving | Program | Avg. |
| TransVTSpotter | LSVTD | 48.2 | 68.2 | 60.3 | 74.2 | 63.8 | 33.7 | 42.1 | 37.9 | 56.5 | 38.1 |
| | RoadText | 5.6 | 4.3 | 3.0 | 9.6 | 6.7 | 0.7 | 3.2 | 2.5 | 4.2 | 3.5 |
| | LSVTD&RoadText | 45.9 | 70.4 | 62.3 | 75.3 | 67.6 | 35.7 | 50.5 | 40.9 | 54.5 | 41.3 |
| | BOVText | **80.7** | **82.3** | **78.1** | **91.3** | **81.7** | **53.3** | **70.4** | **61.0** | **83.1** | **64.7** |

tracking decoder takes the object feature from previous frames as input and predicts the corresponding tracking rotated boxes. Finally, with detection rotated boxes and tracking rotated boxes, TransTT obtains the final tracking result by box IoU matching and the Kuhn-Munkres(KM) algorithm [18]. **Recognizer.** Following MASTER [26] is utilized to predict output sequence with 2D-attention.

## 4 Experimental

In this section, we mainly conduct experiments on our BOVText. More experiments, such as the performance of TransVTSpotter in other datasets, would be provided in the appendix.

### 4.1 Implementation Details

Except for the TransVTSpotter, we also adopt CRNN [38], RARE [39] as the recognition baseline and PSENet [48], EAST [62], DB [22] as the detection baseline to evaluate our BOVText. *Detection*: we train detectors with pre-trained model on COCOText [43]. *Recognition*: the network is pre-trained on the *chinese ocr*[10] and MJSynth [14], then fine-tuned on our BOVText. All of our experiments are conducted on 8 V100 GPUs. AdamW [25] as the optimizer and the batch size is set to be 16. The initial learning rate is 2e-4 for the transformer and 2e-5 for the backbone. The weight decay is 1e-4 All transformer weights are initialized with Xavier-init [10]. The data augmentation includes random horizontal, scale augmentation, resizing the input images whose shorter side is by 480-800 pixels while the longer side is by at most 1333 pixels. The model is first pre-trained on COCOText [43] and then fine-tuned on other video text training sets. For each iteration, two adjacent frames are randomly selected from one video from training set to train our model.

### 4.2 Attribute Experiments Analysis for BOVText

**New Scenarios, New Challenge for Video Text Tasks.** Figure. 5 and Table. 3 gives the tracking performance $ID_{F1}$ of TransVTSpotter in different scenarios of BOVText. Two new insights can be present from the figure and table: 1) The existing benchmark datasets cannot effectively test the effectiveness of advancing algorithm on some novel scenarios (*e.g.,* NewsReport, Cartoon) for first proposed in BOVText. LSVTD [5] and RoadText-1k [32], as the two largest data sets on the existing video text datasets are used to compare with our BOVText. TransVTSpotter achieves a tracking performance $ID_{F1}$ of 70.4% on $NewsReport$ scenario with BOVText training set, around 20 percent point improvement than training with LSVTD [5] and RoadText-1k [32]. We argue that the main cause is that there existing a mass of caption texts in $NewsReport$ scenario, but LSVTD and RoadText almost no such dense text scenario, which is a new challenge for algorithms. Besides, training with only RoadText-1k obtains a low performance no matter which scenarios. There are two main causes for this. Firstly, the location annotation of RoadText-1k is an upright bounding box(two points), but the counterpart of BOVText is multi-orient boxes(four points). Secondly, the data domain is entirely different for the two datasets. Compare with various scenarios (*e.g.,* Game, Sports) and text types (*e.g.,* long caption text, big text), the scenario of RoadText-1k only contains small and low-resolution road signs, plate number on driving scenarios. 2) Huge performance gap existing during different scenarios. As shown in Figure. 5, the model achieves the best performance with a $ID_{F1}$ of 88.4% in $Fishery$ videos, since the conspicuous text instances, simple foreground (*i.e.,* caption texts) and background are designed in cartoon videos. By comparison, several scene categories obtain extremely dissatisfied performance due to complex background, various text appearance, and unsteady camera movements, such as $Sports$ of 46.7% and $Travel$ of 55.8%.

**Bilingual Recognition, New Challenge.** As shown in Figure. 4 (a), the text recognition results for different languages are provided. In summary, the alphanumeric recognition result (about 47%)

---
[10]https://github.com/YCG09/chinese_ocr

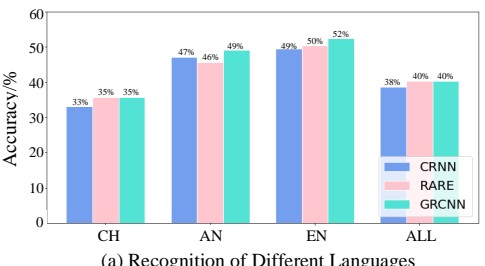

(a) Recognition of Different Languages

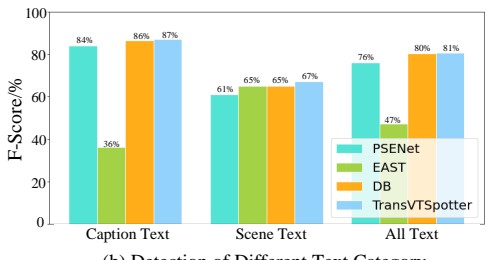

(b) Detection of Different Text Category

Figure 4: **Attribute Experiments of BOVText**. (a) Recognition accuracy of different models in different languages. (b) Detection of different model in caption or scene text. 'CH', 'AN', 'EN' and 'ALL' refer to 'Chinese', 'Alphanumeric', 'English' and 'All Characters', respectively.

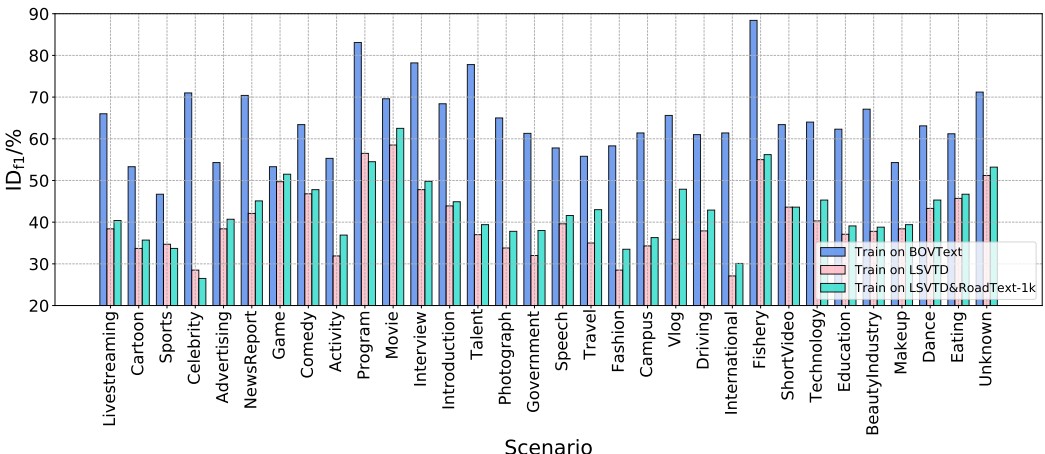

Figure 5: **Tracking performance (*i.e.,* $ID_{F1}$) with TransVTSpotter in different scenarios**. 'Training on LSVTD' and 'Training on BOVText' denotes training on LSVTD train set and BOVText train set, respectively. Huge performance gap existing during different scenarios.

is better than the Chinese recognition result (about $35\%$), regardless of the models. The final results (about $40\%$) for all characters are satisfactory, can not meet the requirement of the application. Unlike English, Chinese contains thousands of characters($3,856$ Chinese characters v.s. $26$ English characters on BOVText), which are difficult to recognize.

**Long Caption Text, New Challenge.** As shown in Figure. 4 (b), for DB [22], PSENet [48] and our TransVTSpotter, the performance of caption text is better than the counterpart of scene text (around $80\%$ vs. $60\%$) due to the more clear and bigger caption text. But for EAST [62], long caption text show a low performance with $36\%$ F-score. The prime reason is that caption texts are all long text ( average width-height ratio: $6.8$ for caption *v.s.* $2.3$ for scene text on BOVText), a different case for EAST [62], as shown in YouTubeDemo. However, the existing video text datasets hardly contain long caption texts, our BOVText can fill out the gap for a more comprehensive evaluation of text types.

### 4.3 Text Detection, Recognition, Tracking and Spotting on BOVText

**Video Text Detection and Recognition.** As shown in Table. 4, image-based text detection on BOVText is not unsatisfactory, with lower results than these methods report on existing text datasets. For example, EAST obtains an f-score of $47.0\%$ compared to the F-score of $80.7\%$ on icdar2015 [63], but our TransVTSpotter obtain an f-score of $81.7\%$ on BOVText, at least $1\%$ improvement compare to the image-based detectors (*i.e.,* DB, PSENet and EAST). For text recognition, CRNN [38] based on CTC loss, RARE [39] with attention mechanism and GRCNN [44] as the base text recognizers to test our BOVText. The text annotation in our BOVText covers two languages (*i.e.,* English and Chinese), thus we conduct several experiments for each language. The recognition model only yields about $40\%$ accuracy on our dataset, but the same model reports at least $90\%$ on most benchmark datasets [17] for text recognition. The main reasons have two points: (1) The proposed BOVText is

Table 4: **Detection and Recognition Performance on BOVText**.

| Detection Performance/% | | | | Recognition Performance/% | | | | | | | | |
|---|---|---|---|---|---|---|---|---|---|---|---|---|
| | | | | | Pretrained | | | | Fine tuned | | | |
| Method | Precision | Recall | F-score | Method | Chinese | Alphanumeric | English | All | Chinese | Alphanumeric | English | All |
| EAST [62] | 55.4 | 40.8 | 47.0 | CRNN [38] | 26.0 | 32.1 | 36.1 | 23.2 | 33.2 | 47.1 | 49.5 | 38.6 |
| PSENet [48] | 78.3 | 75.7 | 77.0 | RARE [39] | 25.2 | 34.2 | 37.4 | 23.5 | 35.6 | 45.7 | 50.4 | 40.2 |
| DB [22] | 84.3 | 77.6 | 80.8 | GRCNN [44] | 23.1 | 39.8 | 40.4 | 26.7 | 35.6 | 49.2 | 52.4 | 40.3 |
| TransVTSpotter | **86.2** | **77.4** | **81.7** | - | 26.2 | 40.3 | 42.1 | 29.1 | 36.2 | 48.9 | 52.1 | 40.4 |

Table 5: **Text Tracking and End to End Video Text Spotting Performance on BOVText.** Text tracking trajectory id generation use a method proposed in [49].

| Method | | Text Tracking on BOVText | | | | | End to End Text Spotting on BOVText | | | | |
|---|---|---|---|---|---|---|---|---|---|---|---|
| Detection | Recognition | $ID_P$/% | $ID_R$/% | $ID_{F1}$/% | MOTA | MOTP | $ID_P$/% | $ID_R$/% | $ID_{F1}$/% | MOTA | MOTP |
| EAST [62] | CRNN [38] | 29.9 | 26.5 | 28.1 | -0.216 | 0.758 | 6.8 | 6.9 | 6.8 | -0.793 | 0.763 |
| | RARE [39] | | | | | | 4.2 | 5.3 | 4.7 | -1.05 | 0.772 |
| PSENet [48] | CRNN [38] | 52.4 | 40.9 | 45.9 | 0.521 | 0.775 | 31.3 | 26.7 | 28.8 | -0.170 | 0.792 |
| | RARE [39] | | | | | | 35.6 | 28.8 | 31.7 | -0.162 | 0.803 |
| DB [22] | CRNN [38] | 55.2 | 42.9 | 48.3 | 0.532 | 0.783 | 38.8 | 30.1 | 33.7 | -0.132 | 0.813 |
| | RARE [39] | | | | | | 41.1 | 29.3 | 34.2 | -0.126 | 0.811 |
| TransVTSpotter(ours) | | **71.0** | **59.7** | **64.7** | **0.682** | **0.821** | **43.6** | **38.4** | **40.8** | **-0.014** | **0.820** |

bilingual, and the category number of Chinese characters in real-world is much larger than those of Latin languages. (2) The video texts are quite blurred, out-of-focus, and the distribution of characters is relatively smaller than the static image counterparts, which presents more challenges.

**Video Text Tracking.** As shown in Table. 5, $ID_{F1}$ (64.7%) of our TransVTSpotter achieves the best performance, at least 10%+ improvement than other methods. Besides, without NMS and other post-processing, TransVTSpotter presents 9 fps no matter which dataset. More details and analysis concerning TransVTSpotter can be obtained in the appendix. And EAST shows the worst performance with a $ID_{F1}$ of 28.1%. The $IDF_1$ of EAST [62] is lower with 17.8% gap than that of PSENet [48]. The main reason is that BOVText contains a mass of long text instances, but regression-based EAST can not deal with the long text cases well. The performance of DB is similar to that of PSENet for both all are the segmentation-based methods.

**End to End Text Spotting in Video**. Detection and text tracking tasks are paving the way for the recognition task. Table. 5 shows the performance of text spotting on BOVText. Similar to the text tracking performance, our TransVTSpotter achieves the state-of-the-art performance with at least 6% $ID_{F1}$ improvement compared to the other methods. Besides, the MOTP of TransVTSpotter achieves 82%, 1% percent points improvement than the counterpart of using DB and RARE. The great performance for 40.8% $ID_{F1}$ and 82% MOTP present satisfactory tracking and recognition trajectory and detection results, respectively. The corresponding performance using EAST [62] as the detector in video text spotting is still not satisfied with around 5% $ID_{F1}$ and $-0.8$ MOTA. Without TransVTSpotter, the combination of DB [22] and RARE [39] achieves the best performance with a 34.2% $ID_{F1}$, but there is at least 6% gap compare to our method.

# 5 Conclusion and Future Work

In this paper, we establish a large-scale, bilingual open-world benchmark dataset for video text tracking and spotting, termed BOVText, with four description information, *i.e.,* , bounding box, tracking ID, recognition content, and text category label. Compare with the existing benchmarks, the proposed BOVText mainly contains four advantages: large-scale training set (*i.e.,* 2,021+video), 32 open real scenarios (*Sportscast, Life Vlog, Game*), bilingual annotation, and abundant text types annotation(Caption, title, and scene text). Besides, we first propose an end-to-end video text spotting framework with Transformer, termed TransVTSpotter, which presents a simple, but efficient attention-based query-key pipeline. On ICDAR2015(video), TransVTSpotter achieves the state-of-the-art performance with faster inference speech.

More importantly, we hope the proposed BOVText and TransVTSpotter would provide a standard benchmark to facilitate the advance of video-and-language research in the future. For example, video captioning with reading comprehension is a novel challenging task requiring models to read text in the video, recognize the video content, and comprehend both modalities jointly to generate a succinct video caption. And more detailed discussions for these content will be provided in the appendix.

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
