# OpenReview forum: "A Bilingual, OpenWorld Video Text Dataset and End-to-end Video Text Spotter with Transformer"
_NeurIPS.cc/2021/Track/Datasets_and_Benchmarks/Round2 — NeurIPS 2021 Datasets and Benchmarks Track (Round 2)_

### Official Review · Reviewer_E92A · 2021-09-20

**Rating:** 7
**Confidence:** 3

**Strengths:**

- A large scale video text spotting datasets that are (1) multilingual,  (2) composed of open-world videos from 30+ categories (many are not covered in existing datasets) and  (3) annotated with rich text type annotations.
- Compared with previous datasets, MOVText is a few magnitude larger and well-annotated.
- The dataset supports more diverse evaluation tasks.
- Abundant analysis on the dataset are presented.
- The authors also showcased its potential to not only be a decent video text spotting benchmark but also to extend to video-and-language tasks.
- The proposed baseline model achieve competitive performance on existing dataset and establishes a strong baseline on the proposed dataset.


**Weaknesses:**

No major weakness. Please refer to the sections below (Clarity/Documentation/Additional Feedback) for some detailed questions I have.

**Additional Feedback:**

About text type `caption text`.
- In real-life applications, what is the use case that we will need to use video-text spotting methods to get the caption texts? Wouldn't ASR be easier?
- In some cases, there could be `caption text` in videos without audio. Would it be more useful to annotate caption texts that are not extractable by ASR differently?

**Clarity:**

Overall, the paper is clear. Below are some small issues or suggestions.
- TransSpotter or TransVTSpotter or Video TransSpotter? Please unify the model name for better clarity.
- Section 5.1 mentioned that TransSpotter is first pre-trained on COCOText then finetuned on video-text datasets, but the other baselines are trained in different ways (L255-257). Then wouldn't it be a unfair comparison between TansSpotter and other models?
- L200, it might be more clear to include one sentence describing which errors can MOT help to pinpoint its source.
- L134, how are the two rounds of screening conducted? with a pre-trained model or by human annotators?

**Correctness:**

I believe the dataset is constructed in a sound way. The evaluation methods and experiment design also seems appropriate and performed correctly to me.

**Documentation:**

Section 3.1 introduces the data collection and organization in details.  A few details are missing or maybe misleading.
- I have noticed that the GitHub repo for dataset release (https://github.com/weijiawu/MOVText-Benchmark) uses Apache License, which is a software license. You might want to double check if this is the suitable license.
- In README, the author mentions that the release is under BSD-3 License, and no permitted commercial use, but the LICENSE file is Apache and permits commercial use. BTW, it seems that BSD-3 is still for software release. (https://en.wikipedia.org/wiki/BSD_licenses)
- Data not available at the GitHub repo yet, but the release plan has already been listed.
- I do have a question about the authors' plan in releasing the video data. How are you planning to release the video data? the raw videos or the extracted video frames? In both cases, there might be copyright issues, especially for the YouTube videos (which Kuaishou mostly probably do not own the copyright). If only YouTube links are provided, there maybe issues when the videos become unavailable, hence it would be hard for future works working on the dataset.

**Ethics:**

Maybe. It depends on how the authors plan to release the videos from YouTube.

**Relation To Prior Work:**

Clearly discussed.
- Section 2 and section 3.2 compares MOVText against previous video-text spotting datasets and highlights MOVText's contribution as (1) large-scale (2) multilingual (3) open-world videos (4) rich text type annotations.
- L59-L65 also compares the proposed TransVTSpotter with previous methods. TransVTSpotter requires less steps hence more efficient while achieving better accuracy.

**Summary And Contributions:**

This paper presents a video text spotting benchmark MOVText that are multilingual, composed of open-world videos from 30+ categories and annotated with rich text type annotations. A simple yet efficient video text spotting model is also introduced, that achieves state-of-the-art performance on ICDAR2015 (video) dataset, and establishes a strong baseline on MOVText.

---

> ### Author Response · Authors · 2021-09-26
> **Response to Reviewer E92A**
>
> Thank the reviewer for the valuable comments and suggestions.
>
> **Q1:** *The licensing and copyright information for the videos data.*
> **A1:** The video data includes two parts: $1,494$ videos from KuaiShou and $356$ videos from YouTube. For those videos from KuaiShou, we mask private information such as the human face, which has passed the examination of the legal department and copyright department of KuaiShou corporation. Thus we own the copyright for these videos. For those videos from YouTube, to the best of our knowledge at the time of download, we have exercised caution to download only those videos that were available on YouTube with a Creative Commmons CC-BY (v3.0) License. We don't own the copyright of those videos and provide them for non-commercial research purposes only. All data in our project is open source under CC-by 4.0 license and only be used for research purposes. We have updated the related information in the revised manuscript(https://github.com/weijiawu/BOVText-Benchmark/blob/main/Dataset/image/Supplementary%20Material.pdf) and our GitHub repo(https://github.com/weijiawu/BOVText-Benchmark)
>
> **Q2:** *The dataset uses Apache License, which is a software license.*
> **A2:** Thank you for pointing this out. We have modified the license to “CC-by 4.0”(https://github.com/weijiawu/BOVText-Benchmark).
>
> **Q3:** *What is the use case that we will need to use video-text spotting methods to get the caption texts? Wouldn't ASR be easier? In some cases, there could be caption text in videos without audio. Would it be more useful to annotate caption texts that are not extractable by ASR differently?*
> **A3:** Video-text spotting is necessary for the caption texts in some cases such as video caption translation and video understanding. The following points are for reference:
>
> - There exists many caption texts or overlap texts without audio, such as songs title, logos, game scores in sports TV.
>
> - In some applications, it is impractical to annotate caption texts directly. For example,  video retrieval application of international video-sharing websites needs to deal with hundreds of millions of videos.
>
> - In fact, the annotation cost of scene text is much higher than the cost of caption texts due to the clear and big caption texts without rotation. Thus, even though ASR shows better performance on those texts with audio, the annotation cost of the dataset still can not save much.
>
>
> **Q4:** *Section 5.1 mentioned that TransSpotter is first pre-trained on COCOText then finetuned on video-text datasets, but the other baselines are trained in different ways (L255-257). Then wouldn't it be an unfair comparison between TansSpotter and other models?*
> **A4:** Sorry about the experiment setting. We will add the results of other baselines using pre-trained model on COCOText for a fair comparison, as shown in the following table.
>
> |  Method | Pre-trained on COCOText | $ID_P/%$| $ID_R/%$| $ID_{F1}/%$| MOTA | MOTP
> |  ----  | ----  |----  |----  |----  |----  |----  |
> | DB&RARE| No | 33.7  | 29.9 | 31.7  |  0.438 |  0.765
> | DB&RARE| Yes | 34.2  | 29.8 | 31.9  |  0.448 |  0.763
> | TransVTSpotter(ours)| No | 65.2  | 51.6 | 57.8  |  0.648 |  0.751
> | TransVTSpotter(ours)| Yes  | 65.6  | 52.2  | 58.2  |  0.682 |  0.771
>
> As shown in the table, we supplement the result of DB&RARE(the best model among other baselines) using pre-trained model. Since the size of the proposed dataset is much larger than COCOText (1,600,000 frame images v.s 30,000 images), thus the performance improvement from the pre-trained model is quite limited, and there is no essential influence for the performance gap between our method and other baselines. We will unify the experimental setting, and updating the results of all models using pre-trained model on COCOText in the revised manuscript.
>
> The main cause for the inconsistent experiment setting is as follows: the experiments of other baselines are done in the early stage(Around May). But our method(TansSpotter) is proposed in the later stage(Around August). We firstly attempt our method on the public benchmark(e.g., ICDAR2015) with the pre-trained model on COCOText, the related experiments are provided in Table 5 and Table 6, supplementary material. And then the same experiment setting is conducted on our dataset without check.
>
> **Q5:** *TransSpotter or TransVTSpotter or Video TransSpotter? Please unify the model name for better clarity.*
> **A5:** Thank you for pointing this out. We have unified the term to "TransVTSpotter" in [the revised version]( https://github.com/weijiawu/BOVText-Benchmark/blob/main/Dataset/image/BOVText.pdf).
>
> **Q6:** *L134, how are the two rounds of screening conducted? with a pre-trained model or by human annotators?*
> **A6:** Sorry to confuse you. We choose the video clips with text by human annotators. Since manual screening shows a better screening result than the pre-trained model, especially for tiny text detection. We will make it clear in the revised version, Line136.

---

> ### Comment · Reviewer_E92A · 2021-10-04
> **Final rating**
>
> Thanks the authors for addressing my concerns. I would like to keep my original rating and inclined to accept this paper.

---

### Official Review · Reviewer_oaux · 2021-09-20

**Rating:** 5
**Confidence:** 4

**Strengths:**

1. Good coverage of related work
2. Good comparison with other datasets
3. MOVText introduces certain challenges which were not posed by other existing datasets

**Weaknesses:**

1. The language is unclear and grammatically incorrect at way too many places making the paper difficult to understand
2. I is not clear what exactly authors mean by "incidental text". Is it same as the scene text? Or is it a generic term for scene text, titles and captions taken together?
3. What is meant by "open" in the 30+ "open categories" and how is it depicted in Figure 1 a?
4. The task of video text spotting should have been first defined the Introduction
5. It is not clear whether the models are trained using each video as a training example or the sub sequences of frames from the videos as training examples? How are these sequences formed?
6. Line 61-65 is not clear
7. Frequent use of incomplete, bullet-point-like sentences has hampered the readability of the text
8. Line 122-124: Please elaborate how the temporal processing is helping the task of text spotting?
9. It is not clear from the paper about the license or copy right information of the videos downloaded from YouTube and KuaiShou. Also, what exactly is a "text video class"? Does it mean those videos which have a lot of text in them in addition to the captions?
10. Line 134: what do you mean by "ordinary videos" here?
11. The caption of Fig 3 needs to be re-written to properly highlight the four components
12. Is the dataset actually multi-lingual or bi-lingual with English and Chinese as the two languages for text?


**Additional Feedback:**

Captured in "Weaknesses" above.

**Clarity:**

No. The language is unclear and grammatically incorrect at way too many places making the paper difficult to understand. Frequent use of incomplete, bullet-point-like sentences has also hampered the readability of the text. The organization of Section 5 must be improved to better highlight the impact.

**Correctness:**

The details seem to be correct except for my reservations due to the points mentioned under "weaknesses" above.

**Documentation:**

1. No mention of availability and maintenance in the main text.
2. The licensing information of the videos downloaded is not clear
3. Lack of details on data organization

**Ethics:**

Nothing apparent.

**Relation To Prior Work:**

Yes.

**Summary And Contributions:**

The authors propose a new large scale dataset (MOVText) for video text spotting. Because of rich annotations the dataset lends itself well to other video text tasks like detection, recognition and tracking as well. It is multi lingual and covers various open-world scenarios. The authors also propose a transformer based method for end to end video text spotting which achieves state of the art performance on MOVText.

---

> ### Author Response · Authors · 2021-09-26
> **Response to Reviewer oaux (Part1)**
>
> Thank the reviewer for the valuable comments and suggestions.
>
> **Q1:** *what exactly authors mean by "incidental text"?*
> **A1:** Sorry to confuse you. The "incidental text", as a generic term in the scene text field, firstly is proposed on ICDAR2015[1]. The simple definition for "incidental" has been introduced in the caption of Table 1. Incidental scene text refers to text that appears indoor and outdoor scenarios(*e.g., walking outdoors, driving*) in daily life without the user having taken any specific prior action to cause its appearance or improve its positioning/quality in the frame.
>
> **Q2:** *What is meant by "open" in the 30+ "open categories" and how is it depicted in Figure 1 a?*
> **A2:** "Open" is defined as any scenarios without region limitation, and includes abundant virtual scenarios, *e.g., Game, Sport(NBA)* in our paper (Caption of Table 1). Figure 1 shows that our dataset is collected from the worldwide user of YouTube and Kuaishou, cover various daily scenarios without region limitation and virtual scenes. But the previous video text datasets usually are collected toward a special city or language from the hand-held camcorder. We have refined it in the revised version, Line52-56.
>
> **Q3:** *The task of video text spotting should have been first defined the Introduction*
> **A3:** Thank you for the valuable suggestion. We have added the definition "*Video text spotting(VTS) is the task that requires simultaneously detecting, tracking and recognizing text instances in a video sequence.*" to the Introduction Line28-29 in the revised version. The updated manuscript can be found in https://github.com/weijiawu/BOVText-Benchmark/blob/main/Dataset/image/BOVText.pdf.
>
> **Q4:** *Whether the models are trained using each video as a training example or the sub sequences of frames from the videos as training examples? How are these sequences formed?*
> **A4:** We don't understand what's meaning of the question. We guess that the reviewer is confused about how to train the model in each iteration, and how the input sequence frames formed. The models are trained using the sub sequences of frames as the input. For each iteration, two consecutive frames are randomly selected from one video to train our model. For example, $F_i$($i\mbox{-}th$ frames in one video) and $F_{i+k}$($k<5$), as the consecutive frames are taken to input the model. "k" is used to augment the diversity of data among a reasonable range. We will make it clear in the revised version, Line270-271.
>
> **Q5:** *Line 61-65 is not clear.*
> **A5:** The proposed TransVTSpotter includes two steps. 1) Text tracking: for each consecutive frame image, we obtain the tracking trajectories of text by box IoU matching between the predicted detection box[4] and the predicted tracking box[3], where the detection box are obtained by taking an object query as input, just like DETR[4]. And features from previously frames to form another “track query” to discover associated objects(*i.e.,* the predicted tracking boxes) on the current frames. 2) Text recognition: the tracked texts will be recognized with the attention-based text recognizer[5]. Thank you for the valuable suggestion, we have refined it in the updated version, Line64-71. The updated manuscript can be found in https://github.com/weijiawu/BOVText-Benchmark/blob/main/Dataset/image/BOVText.pdf.
>
> **Q6:** *Line 122-124: Please elaborate how the temporal processing is helping the task of video text spotting?*
> **A6:** Video text spotting is the task that requires simultaneously detecting, tracking, and recognizing text instances in a video sequence. The great temporal processing could help better associating and tracking text with the same identification. For example, design an end-to-end tracker to track already detected text box from the previous frame, that's the content of our work. Besides, many previous works[6][7] have been proved the idea.
>
> **Q7:** *The licensing information of the videos downloaded is not clear.*
> **A7:** We have added one section to state the license and copyright in the revised Supplementary Material, Line163. The updated supplementary material can be found in https://github.com/weijiawu/BOVText-Benchmark/blob/main/Dataset/image/Supplementary%20Material.pdf.

---

> ### Author Response · Authors · 2021-09-26
> **Response to Reviewer oaux (Part2)**
>
> **Q8:** *What exactly is a "text video class"? Does it mean those videos which have a lot of text in them in addition to the captions?*
> **A8:** There is various daily life scenarios with a variety of text types, such as *the texts of the package in a supermarket, road sign on the road, caption and logo on movie*. "text video class" denotes different daily scenarios with different text types. The concept of the term is proposed in the 2015 13th International Conference on Document Analysis and Recognition(ICDAR)[1], where firstly proposed seven scenarios such as *"Browse products in a supermarket, Search for a location in a building, Driving"*. In our paper, we propose 32 open classes with more extensive semantics such as *"driving, travel, cartoon, movie, game"*. We have revised the term "text video class" to "different scenarios with text" to avoid confusion, Line134-135 in the revised version.
>
> **Q9:** *Line 134: what do you mean by "ordinary videos" here?*
> **A9:** Sorry to confuse you. "ordinary videos" refer to those videos without scene text and caption text. We have refined it in the revised version, Line139-140.
>
> **Q10:** *The caption of Fig 3 needs to be re-written to properly highlight the four components*
> **A10:** Thank you for the valuable suggestion. We have refined it, and more details please see the revised version in https://github.com/weijiawu/BOVText-Benchmark/blob/main/Dataset/image/BOVText.pdf.
>
> **Q11:** *Is the dataset actually multi-lingual or bi-lingual with English and Chinese as the two languages for text?*
> **A11:** The current dataset version supports bi-lingual with English and Chinese. We have revised all "multi-lingual" to "bi-lingual" for a more accurate description in the updated version. And the related descriptions on Github(https://github.com/weijiawu/BOVText-Benchmark) and supplementary material have also been updated.
>
> [1] Karatzas, Dimosthenis, Lluis Gomez-Bigorda, Anguelos Nicolaou, Suman Ghosh, Andrew Bagdanov, Masakazu Iwamura, Jiri Matas et al. "ICDAR 2015 competition on robust reading." In 2015 13th ICDAR, pp. 1156-1160. IEEE, 2015.
>
> [2] Bradski, Gary, and Adrian Kaehler. Learning OpenCV: Computer vision with the OpenCV library. " O'Reilly Media, Inc.", 2008.
>
> [3] Sun, Peize, Yi Jiang, Rufeng Zhang, Enze Xie, Jinkun Cao, Xinting Hu, Tao Kong, Zehuan Yuan, Changhu Wang, and Ping Luo. "Transtrack: Multiple-object tracking with transformer." arXiv preprint arXiv:2012.15460 (2020).
>
> [4] Carion, Nicolas, Francisco Massa, Gabriel Synnaeve, Nicolas Usunier, Alexander Kirillov, and Sergey Zagoruyko. "End-to-end object detection with transformers." In European Conference on Computer Vision, pp. 213-229. Springer, Cham, 2020.
>
> [5] Ning Lu,Wenwen Yu, Xianbiao Qi, Yihao Chen, Ping Gong, Rong Xiao, and Xiang Bai. Master: Multi-aspect non-local network for scene text recognition. Pattern Recognition, 117:107980, 2021.
>
> [6] Wang, Ning, Wengang Zhou, Jie Wang, and Houqiang Li. "Transformer Meets Tracker: Exploiting Temporal Context for Robust Visual Tracking." In CVPR, pp. 1571-1580. 2021.
>
> [7] Wang, Yuqing, Zhaoliang Xu, Xinlong Wang, Chunhua Shen, Baoshan Cheng, Hao Shen, and Huaxia Xia. "End-to-end video instance segmentation with transformers." In CVPR, pp. 8741-8750. 2021.

---

> ### Author Response · Authors · 2021-09-26
> **Response to Reviewer oaux (Part3)**
>
> **Q12:** *Lack of details on data organization.*
> **A12:** We have added the related information in our Github(https://github.com/weijiawu/BOVText-Benchmark).
>
> **Q13:** *No mention of availability and maintenance in the main text.*
> **A13:** We have added one statement concerning the hosting and maintenance plan in our Github(https://github.com/weijiawu/BOVText-Benchmark). The following are for reference.
>
> - **Clear maintenance plan**. We open and establish the GitHub repository three months ago. During this period, we mainly have done four tasks: 1) writing and refining my manuscript. 2) dealing with the privacy information of the data and waiting for the legal department review of Kuaishou Technology. 3) finishing experiments of the related baselines on public benchmark or our dataset, and releasing the code for TransVTSpotter(https://github.com/weijiawu/TransVTSpotter). 4) establishing a  GitHub repository to maintain the basic information(Description, Tasks, and Metrics). In the next two months, there is a clear maintenance plan for us: 1) merging and releasing the whole dataset after further review. 2) updating evaluation guidance and script code for four tasks(detection, tracking, recognition, and spotting). 3) Hope to host a competition concerning our work for promotional and publicity.
>
> - **We promise that the dataset and website will be maintained for at least two years and the dataset download link is always valid**. Weijia Wu, as the first author, is a Ph.D. student focusing on video-and-language research topic at Zhejiang University(ZJU) since 2018 and will graduate at least after 2023 year. At least for the next two years, the author still plays an active participant in the video text field and maintaining the dataset. Besides, since there are few video-based text spotting methods and no one is open source, we hope the proposed dataset and method to become a standard benchmark for promoting video text spotting in the community. To achieve the goal, we will maintain and continually updating the project and Github repository.
>
> - **The dataset will supports more video-and-language tasks in the future**. As the presentation in the section "Link to Other Video-and-Language Applications", Supplementary Material, more video-and-language tasks such as text-based video retrieval task will be supported by our dataset. Video retrieval with textual cues[1] is also a very important application direction for video-and-language research. To the best of my knowledge, video retrieval with text information in the video is still almost a blank field of study and immature application in the industry. The proposed BOVText with various text types (e.g., caption, song title, logos, street signs, business signs) and annotation can promote the research concerning efficient video retrieval. Therefore, the long time maintenance is inevitable and necessary to support more video-and-language tasks in the future.
>
> [1] Anand Mishra, Karteek Alahari, and CV Jawahar. Image retrieval using textual cues. In Proceedings of the IEEE International Conference on Computer Vision, pages 3040–3047, 2013.

---

### Official Review · Reviewer_q1aQ · 2021-09-21
**A large-scale open-world video text dataset**

**Rating:** 6
**Confidence:** 4
**Correctness:** The claims made in the submission are…
**Clarity:** The paper is well written.

**Strengths:**

1. The open-world large-scale video text dataset has not been proposed before. It could potentially inspire and benefit future work in this direction.
2. The authors provide detailed information and analysis on the dataset, including the data collection and annotation process, data statistics, and evaluation tasks and metrics. Experiments are conducted on different methods and different tasks.
3. This paper is clearly written and easy to follow.

**Weaknesses:**

1. Some of the dataset statistics are missing. For example, how many types of language are there in the dataset? What are the percentages of each language? What are the percentages of each scenario?
2. As the dataset paper, the method part is a bit unnecessary. I would suggest the authors to put less weight on the proposed method and more emphasis on the dataset and benchmarking previous methods and baselines.

**Additional Feedback:**

N/A.

**Documentation:**

There is sufficient detail on data collection and annotation. Data statistics can be more complete (see weakness). The authors currently released a small set of the dataset (46 videos) and promise to release the whole dataset by Oct 15. Maintainance and ethical and responsible use are not mentioned. Hosting and maintenance plan are missing. The authors have a TODO list on the dataset repo, and it remains to be completed.

**Ethics:**

No.

**Relation To Prior Work:**

It clearly discussed how this work differs from previous datasets in this field.

**Summary And Contributions:**

This paper proposes a large-scale open-world video text dataset (MOVText). Compared with previous datasets, it contains more frames and more scenarios. It also contains text types annotation and multilingual text annotation. This dataset can be used for multiple text video tasks including text detection, text recognition, video text tracking, and end-to-end text spotting in Videos. The authors provide detailed information on the data collection and data annotation process, as well as the dataset statistics and comparison with previous datasets. This paper also proposes an end-to-end text spotting framework named TransVTSpotter. Experiments on different tasks are conducted and the results of different methods are reported.

---

> ### Author Response · Authors · 2021-09-26
> **Response to Reviewer q1aQ (Part1)**
>
> Thank the reviewer for the valuable comments and suggestions.
>
> **Q1:** *Some of the dataset statistics are missing. For example, how many types of language are there in the dataset? What are the percentages of each language? What are the percentages of each scenario?*
> **A1:** In fact, we have provided the related dataset statistics. Statistics of text language and category are provided in Table 2, Supplementary Material. The Data Distribution for each scenario is provided in Table 1 and Figure 1, Supplementary Material. To further highlight these statistics, we present the related information as the following:
>
> - **The data distribution for 32 open scenarios.** As shown in the [Table](https://github.com/weijiawu/BOVText-Benchmark/blob/main/Dataset/image/table1.png) and [Figure](https://github.com/weijiawu/BOVText-Benchmark/blob/main/Dataset/image/figure1.png), we present the 31 open scenarios and an "Unknown" scenarios distribution on our dataset in three levels, i.e., video, video frames, and text instances.
>
> - **Statistics of text language and category.** As shown in the [Table](https://github.com/weijiawu/BOVText-Benchmark/blob/main/Dataset/image/table2.png), we provide two language text annotation and four text categories annotation (*i.e., caption, title, scene text, or others*).
>
> Besides,  more details concerning the proposed dataset are provided in the Supplementary Material, such as the details for how to blur human faces.
>
>
> **Q2:** *As the dataset paper, the method part is a bit unnecessary. I would suggest the authors put less weight on the proposed method and more emphasis on the dataset and benchmarking previous methods and baselines.*
> **A2:** Good suggestion! But we actually have put emphasis on the dataset with many details, such as four baselines, three new insights, and comprehensive dataset statistics. In the current version, since we want to highlight the contribution of dataset and method at the same time on the limited page, we chose to put some important dataset comparisons and statistics to the Supplementary Material.
>
> We will attempt to refine the structure of the paper and put more weight on the dataset in the revised version. For example, we will move the dataset statistics of scenarios distribution from the supplementary material to the body manuscript and reduce the content of the method description to one page. The updated manuscript for more emphasis on the dataset can be found in https://github.com/weijiawu/BOVText-Benchmark/blob/main/Dataset/image/BOVText.pdf.

---

> ### Author Response · Authors · 2021-09-26
> **Response to Reviewer q1aQ (Part2)**
>
> **Q3:** *Hosting and maintenance plan are missing.*
> **A3:** We have added one statement concerning the hosting and maintenance plan in our Github(https://github.com/weijiawu/BOVText-Benchmark). The following are for reference.
>
> - **Clear maintenance plan**. We open and establish the GitHub repository three months ago. During this period, we mainly have done four tasks: 1) writing and refining my manuscript. 2) dealing with the privacy information of the data and waiting for the legal department review of Kuaishou Technology. 3) finishing experiments of the related baselines on public benchmark or our dataset, and releasing the code for TransVTSpotter(https://github.com/weijiawu/TransVTSpotter). 4) establishing a  GitHub repository to maintain the basic information(Description, Tasks, and Metrics). In the next two months, there is a clear maintenance plan for us: 1) merging and releasing the whole dataset after further review. 2) updating evaluation guidance and script code for four tasks(detection, tracking, recognition, and spotting). 3) Hope to host a competition concerning our work for promotional and publicity.
>
> - **We promise that the dataset and website will be maintained for at least two years and the dataset download link is always valid**. Weijia Wu, as the first author, is a Ph.D. student focusing on video-and-language research topic at Zhejiang University(ZJU) since 2018 and will graduate at least after 2023 year. At least for the next two years, the author still plays an active participant in the video text field and maintaining the dataset. Besides, since there are few video-based text spotting methods and no one is open source, we hope the proposed dataset and method to become a standard benchmark for promoting video text spotting in the community. To achieve the goal, we will maintain and continually updating the project and Github repository.
>
> - **The dataset will supports more video-and-language tasks in the future**. As the presentation in the section "Link to Other Video-and-Language Applications", Supplementary Material, more video-and-language tasks such as text-based video retrieval task will be supported by our dataset. Video retrieval with textual cues[1] is also a very important application direction for video-and-language research. To the best of my knowledge, video retrieval with text information in the video is still almost a blank field of study and immature application in the industry. The proposed BOVText with various text types (e.g., caption, song title, logos, street signs, business signs) and annotation can promote the research concerning efficient video retrieval. Therefore, the long time maintenance is inevitable and necessary to support more video-and-language tasks in the future.
>
> [1] Anand Mishra, Karteek Alahari, and CV Jawahar. Image retrieval using textual cues. In Proceedings of the IEEE International Conference on Computer Vision, pages 3040–3047, 2013.

---

> ### Comment · Reviewer_q1aQ · 2021-10-04
> **Update after rebuttal**
>
> Thank the authors for providing details on the dataset and maintenance plan in the rebuttal. I incline to accepting this paper after reading the rebuttal.

---

### Official Review · Reviewer_dSbN · 2021-09-22

**Rating:** 6
**Confidence:** 3
**Correctness:** I don't find severe issues in constru…
**Clarity:** Paper is well written.

**Strengths:**

- large scale dataset
- include multiple languages, which promotes diversity.
- new end-to-end transformer model inspired by DETR as a strong baseline


**Weaknesses:**

- only has 2 languages (English, Chinese)

In L58, Arabic is mentioned but I don't find anywhere discussing it so I assume this dataset is Chinese+English. To claim this as a multilingual dataset, I would suggest including more languages, especially those with non-Latin characters



**Additional Feedback:**

N/A

**Documentation:**

Dataset details is documented and a website is provided.

**Ethics:**

I don't find any ethical concerns in this submission.

**Relation To Prior Work:**

Related works are properly discussed.

**Summary And Contributions:**

This paper presents a new text spotting dataset, which is the largest in scale to-date. Furthermore, a new end-to-end transformer model is proposed for video text spotting. This dataset contains English and Chinese.

---

> ### Author Response · Authors · 2021-09-25
> **Author response to the review based on the feedback**
>
> Thank the reviewer for the valuable comments and suggestions.
>
> **Q1:** *In L58, Arabic is mentioned but I don't find anywhere discussing it so I assume this dataset is Chinese+English. To claim this as a multilingual dataset, I would suggest including more languages, especially those with non-Latin characters*
>
> **A1:** Thank you for the valuable suggestion. "Arabic" is a clerical error when we want to express "Arabic Number". The current dataset version supports bi-lingual with English and Chinese. Thus we have removed the "Arabic" and revise "multi-lingual" to "bi-lingual" for a more accurate description in the updated version. The updated manuscript can be found in https://github.com/weijiawu/BOVText-Benchmark/blob/main/Dataset/image/BOVText.pdf. And the related description in our Github(https://github.com/weijiawu/BOVText-Benchmark) has been revised from "multi-lingual" to "bi-lingual".
>
> Due to annotation cost and time cost, the current dataset does not support
> Arabic, but the cost and annotation technology are similar to English and Chinese. More languages could be supported for our dataset in the future. Kuaishou and YouTube, as international video-sharing websites, have the potential to promote multilingual research.

---

### Author Response · Authors · 2021-09-26
**Thank you! Revised Version uploaded.**

We would like to thank our reviewers for their encouraging feedback and thought-provoking comments!

Following up on the reviewers' feedback, we have uploaded a revised version of our paper, extending some sections to include:

- Revising confusing terms. For example, we have revised the title from "A Multilingual, OpenWorld Video Text Dataset" to "A Bilingual, OpenWorld Video Text Dataset".

- Refining the structure of the paper and put more weight on the dataset in the revised version. For example, we move the dataset statistics of scenarios distribution from the supplementary material to the body manuscript.

- Maintenance plan, data organization, and licensing information are supplemented in our Github(https://github.com/weijiawu/BOVText-Benchmark).

---

### Decision · Program_Chairs · 2021-10-11

**Decision:**

Accept

**Comment:**

The reviews are mostly positive. The dataset is much larger than existing ones and is likely to be a valuable resource. The authors have addressed the concerns satisfactorily. I am happy to recommend acceptance.